# The Genetic Complexity of Type-IV Trichome Development Reveals the Steps towards an Insect-Resistant Tomato

**DOI:** 10.3390/plants11101309

**Published:** 2022-05-14

**Authors:** Eloisa Vendemiatti, Rodrigo Therezan, Mateus H. Vicente, Maísa de Siqueira Pinto, Nick Bergau, Lina Yang, Walter Fernando Bernardi, Severino M. de Alencar, Agustin Zsögön, Alain Tissier, Vagner A. Benedito, Lázaro E. P. Peres

**Affiliations:** 1Department of Biological Sciences, Escola Superior de Agricultura “Luiz de Queiroz”, Universidade de São Paulo, Piracicaba 13418-900, Brazil; eloisa.vendemiatti@mail.wvu.edu (E.V.); rodrigotfreitas@hotmail.com (R.T.); mhvicente21@gmail.com (M.H.V.); maisadesiqueira@gmail.com (M.d.S.P.); wfbernardi@yahoo.com.br (W.F.B.); 2Division of Plant & Soil Sciences, West Virginia University, Morgantown, WV 26506, USA; lyang2@mix.wvu.edu; 3Department of Cell and Metabolic Biology, Leibniz Institute of Plant Biochemistry, 06120 Halle (Saale), Germany; nick_89@gmx.net (N.B.); alain.tissier@ipb-halle.de (A.T.); 4Department of Agri-Food Industry, Food and Nutrition, Escola Superior de Agricultura “Luiz de Queiroz”, Universidade de São Paulo, Piracicaba 13418-900, Brazil; smalencar@usp.br; 5Department of Plant Biology, Universidade Federal de Viçosa, Viçosa 36570-900, Brazil; agustin.zsogon@ufv.br

**Keywords:** acylsugars, glandular trichomes, herbivory, *Solanum galapagense*, *Solanum lycopersicum*, whiteflies

## Abstract

The leaves of the wild tomato *Solanum* *galapagense* harbor type-IV glandular trichomes (GT) that produce high levels of acylsugars (AS), conferring insect resistance. Conversely, domesticated tomatoes (*S. lycopersicum*) lack type-IV trichomes on the leaves of mature plants, preventing high AS production, thus rendering the plants more vulnerable to insect predation. We hypothesized that cultivated tomatoes engineered to harbor type-IV trichomes on the leaves of adult plants could be insect-resistant. We introgressed the genetic determinants controlling type-IV trichome development from *S.* *galapagense* into cv. Micro-Tom (MT) and created a line named “*Galapagos-enhanced trichomes*” (MT-*Get*). Mapping-by-sequencing revealed that five chromosomal regions of *S. galapagense* were present in MT-*Get*. Further genetic mapping showed that *S. galapagense* alleles in chromosomes 1, 2, and 3 were sufficient for the presence of type-IV trichomes on adult organs but at lower densities. Metabolic and gene expression analyses demonstrated that type-IV trichome density was not accompanied by the AS production and exudation in MT-*Get*. Although the plants produce a significant amount of acylsugars, those are still not enough to make them resistant to whiteflies. We demonstrate that type-IV glandular trichome development is insufficient for high AS accumulation. The results from our study provided additional insights into the steps necessary for breeding an insect-resistant tomato.

## 1. Introduction

Glandular trichomes (GT) have attracted considerable attention due to their potential as sources of a vast array of specialized plant metabolites [1,2]. Among such metabolites, many have industrial or medicinal value [3,4], whereas others are especially relevant for plant defense against insect pests [5,6]. Domesticated tomato and its wild relatives vary in trichome type, size, and number. Eight morphologically distinct types were defined for the *Lycopersicon* clade (the cultivated tomato and its 16 closest wild relatives), four of which are glandular: types I, IV, VI, and VII [6,7]. Type-IV trichomes (and, to a lesser extent, type-I trichomes, which are rare on tomato leaves) are sources of specialized metabolites called acylsugars (AS) [8,9,10,11,12].

AS molecules consist of aliphatic acyl groups of variable chain lengths (C2 to C12) esterified to a glucose (G) or sucrose (S) moiety at four possible positions [10,13,14]. For instance, S4:23 (2,4,5,12) is a sucrose-based AS esterified with C2, C4, C5, and C12 acyl groups, whose sum of aliphatic carbon atoms is 23. In *Solanum*, AS confer resistance to fungal pathogens [15] and multiple insect pests [16,17,18], such as whiteflies (*Bemisia* spp. Quaintance & Baker; Hemiptera, Aleyrodidae), which are major pests of tomato (*S. lycopersicum* L., Solanaceae) worldwide [19]. AS deter insect and other arthropod attacks via distinct mechanisms, such as poisoning, sticking, and even “tagging” insects to increase prey recognition by predators [20]. An event of horizontal gene transfer from plants to whiteflies was recently demonstrated, which the insects use to neutralize plant-derived toxins [21]. This mechanism is unlikely to evolve for AS detoxification since these compounds act chemically and mechanically to combat whiteflies. Therefore, AS comprise a robust and stable mechanism for tomato’s resistance to whiteflies.

Type-IV trichomes are elongated multicellular structures with a single flat basal cell and a short, uniseriate two- or three-celled stalk (0.2–0.4 mm) topped by a spherical gland. They are particularly abundant in the wild relatives of tomato *S. galapagense* S. C. Darwin & Peralta, *S. habrochaites* S. Knapp & D.M. Spooner, and *S. pennellii* Correll [7,22,23]. These species produce much higher AS amounts than cultivated tomato plants [24,25], which underlies their robust and multiple pest resistances [13,26,27,28,29]. It had long been thought that the domesticated tomato does not develop type-IV trichomes [6,7,22,30]. However, we demonstrated that they appear in the early, juvenile stage of plant development [31]. In the model tomato cv. Micro-Tom (MT), the leaf developmental sequence from the bottom to the top consists of a pair of embryonic leaves (cotyledons), a pair of juvenile leaves (first and second true leaves), and adult leaves (third and upwards). In this accession, type-IV trichomes are found from the cotyledons up to the third leaf, and their presence can be used to track the transition from the juvenile to the adult phase in the tomato life cycle [31]. The lack of type-IV trichomes in the adult phase of domesticated tomatoes is associated with a low AS content and, consequently, susceptibility to herbivores.

Interestingly, the genes coding for the enzymes of the AS biosynthesis pathway are present in the *S. lycopersicum* genome [11,12,13,32]. Indeed, genes coding for four acylsugar acyltransferases (ASAT1–ASAT4) belonging to the BAHD acyltransferase superfamily were identified in the genomes of the domesticated and wild tomato species [11,12,13,14]. Biochemical characterization revealed that these ASATs catalyze the sequential esterification of acyl chains at different positions of a sugar core to generate an acylated structure. Despite being composed of common moieties, AS encompass diverse structures in the Solanaceae. This diversity can be partially explained by the gene duplication and neofunctionalization of ASAT enzymes, which allowed catalytic activities with different substrates (e.g., sucrose, glucose, inositol) [9,12,14,33,34]. Within the same genotype, AS encompass a mélange of molecular structures, possibly resulting from enzyme promiscuity and availability of different acyl chain substrates [14].

These observations led to the hypothesis that a domesticated tomato genotype harboring type-IV trichomes on its adult leaves would accumulate high AS levels and be naturally resistant to insects. We successfully introgressed the genetic determinants underlying type-IV trichome formation on adult leaves from *S. galapagense* LA1401 into *S. lycopersicum* cv MT. The novel MT genotype was named “*Galapagos-enhanced trichomes*” (MT-*Get*) and showed a high density of type-IV trichomes on all leaves throughout plant development. The *S. galapagense* regions introgressed into MT-*Get* were determined by mapping-by-sequencing. This unique genetic material allowed us to investigate the functionality of type-IV trichomes on domesticated tomato and their impact on insect resistance. To this end, we performed RT-qPCR analysis of *ASAT1–ASAT4* transcription in leaf samples and visualized GFP expression under the *ASAT* (*SlAT2*) promoter in type-IV trichome glands. The AS profile of MT-Get was determined using liquid and gas chromatography (LC-MS/MS and GC-MS). A preliminary insect resistance assay was performed using whiteflies (*Bemisia* spp.), one of the main insect pests of tomato and a known target of AS. We found that the presence of type-IV trichomes was not associated with increased AS production and was thus insufficient to confer whitefly resistance in MT-*Get*. However, our findings chart a course for molecular breeding of commercial tomato varieties by stacking of all the necessary genetic components required for resistance to whitefly and other arthropod herbivores.

## 2. Results

### 2.1. Introgression of the Genetic Determinants Controlling Type-IV Trichome Development on Adult Leaves from S. galapagense LA1401 into Tomato (S. lycopersicum cv. Micro-Tom)

*Solanum galapagense* LA1401 was chosen as a source of type-IV trichomes for genetic introgression. This accession is closely related to the domesticated tomato but, unlike it, has a high density of type-IV trichomes on adult leaves, especially on the abaxial surface (Figure 1A,B,D). We have previously shown that domesticated tomato plants bear type-IV trichomes only on the cotyledons and juvenile leaves [31]. We, therefore, set out to introgress the genetic determinants controlling type-IV trichomes development on the adult leaves from *S. galapagense* into the domesticated tomato (*S. lycopersicum* cv. Micro-Tom). The introgression process (Figure 1C) was initiated by crossing *S. galapagense* (pollen donor) with MT (pollen receiver) to produce F_1_ plants. During the trichome characterization of F_1_ plants, we observed a lower density of type-IV trichomes on both leaf sides than the parental *S. galapagense* (Figure 1D,E), suggesting that this trait is dominant or semi-dominant with quantitative components. After the self-fertilization of F_1_ plants, the resulting F_2_ plants were selected for the presence of type-IV trichomes on leaves from the adult phase of plant development. These plants were backcrossed (BC_1_) using MT as the recurrent parent. The process was repeated five more times until a stable BC_6_F_n_ line no longer segregated for the trait was obtained. This new genotype was called “*Galapagos-enhanced trichomes*” (MT-*Get*).

We confirmed the identity of type-IV trichomes on adult leaves of MT-*Get* using scanning electron microscopy (Figure 2). The type-IV trichome is up to 0.4 mm long, cone-shaped, uniseriate, harboring a multicellular stalk with a small glandular cell at the tip and a unicellular and flat base [6,7,35]. This description fits the structures shown in Figure 2C, confirming the presence of type-IV trichomes on leaves of adult MT-*Get* plants. The type-IV trichomes of MT-*Get* have similar morphology and dimensions to that of *S. galapagense* (Figure 2D).

We next verified whether type-IV trichomes of MT-*Get* expressed genes of the AS biosynthesis pathway. Transgenic MT and MT-*Get* plants harboring the *GFP* gene under the control of type-IV/I-specific *SlAT2* promoter [13] were generated. Both MT and MT-*Get* cotyledons, and adult leaves of MT-*Get* plants, displayed type-IV trichomes expressing GFP (Figure 3). Accordingly, the absence of visible GFP signal on adult leaves of the MT plant correlated with the absence of type-IV trichomes (Figure 3E,F). No GFP signal was detected in non-transgenic type-IV trichomes on the leaves of the MT-*Get* control (Appendix A).

### 2.2. Phenotypic Characterization and Genetic Mapping of the “Galapagos-enhanced trichomes” (MT-Get) Introgression Line

To further characterize MT-*Get* plants, we first determined trichome classes and their respective densities on their adult leaves (fifth leaf). An inverse ratio between type-IV and -V trichomes on both leaf sides was observed for MT-*Get* and MT (Figure 4). Type-IV trichomes dominated in MT-*Get*, as opposed to type-V density in MT, on leaves of both juvenile and adult plants (Figure 4 and Appendix A). As expected, type-IV trichomes were not found on adult leaves of MT (Appendix A), while MT-*Get* lacked type-V trichomes on juvenile leaves (Appendix A). We had previously found the inverse ratio between types IV and V trichomes in several other tomato cultivars [31]. Compared to *S. galapagense*, MT-*Get* displayed around three-fold (abaxial) and six-fold (adaxial) fewer type-IV trichomes (Figure 1D and Figure 4). MT-*Get* harbored trichomes types I, III, VI, and VII (Appendix A) in addition to IV and V (Figure 4). *S. galapagense*, on the other hand, showed less trichome diversity than MT-*Get*, with only trichomes types I, IV, and VI found on its leaves (Figure 1D and Appendix A).

We next used mapping-by-sequencing to determine the genetic configuration of MT-*Get*, i.e., the *S. galapagense* genome regions and alleles that were introgressed [36]. We bulk-sequenced the genomes of different phenotype categories of a segregating population to identify common loci responsible for a trait by identifying distinct allelic frequencies between groups. MT-*Get* has a complex genetic composition: discrete *S. galapagense* genome segments were found on the long arms of chromosomes 1, 2, and 3, the short arm of chromosome 5, and a large pericentromeric region of chromosome 6 (Figure 5). The genomic coordinates of the genetic variation from *S. galapagense* present at high frequencies (≥0.8) in the *Get*-like phenotype group of the MT-*Get* segregating population are provided in Appendix A. All the regions present on chromosomes 1, 2, 3, 5, and 6, or smaller combinations thereof, may thus be involved in type-IV trichome formation. To further dissect this complexity, we created sub-lineages of MT-*Get* by screening a segregating population with CAPS markers (Appendix A) that cover the extension of the fragments from *S. galapagense*. We identified three sub-lines of MT-*Get* harboring only the *S. galapagense* segments on chromosomes 1, 2, or 3. Preliminary results revealed that these fragments are involved in the type-IV trichome developmental pathway (Appendix A). When these three genomic fragments are isolated in individual lines (Appendix A), type-IV trichome densities are lower than in MT-*Get,* which carries all fragments (Appendix A). The sequencing results suggest that the *Get* trait has a polygenic basis, involving epistatic interactions among multiple genes on several genomic segments derived from *S. galapagense*. The combinations of these multiple alleles apparently control the persistence and density of type-IV trichomes throughout plant development.

We also verified whether known alleles from *S. galapagense* in the introgressed segments could contribute to other developmental differences independent of trichome traits by comparing the distinct genomic regions between MT-*Get* and MT. Notably, within the chromosome 3 segment, MT-*Get* has *S. galapagense* alleles for *EJ-2* [43] and *FW3.2* [44] (Figure 5B). The pleiotropic effects of non-domesticated alleles of the *FW3.2* gene, which codes for a *P450 monooxygenase*, lead to a reduction in fruit mass and increased shoot branching [44]. Both phenotypes are present in MT-*Get* (Appendix A), which harbors the *S. galapagense fw3.2* allele. Another gene controlling fruit mass is *FW2.2* [45], although it cannot be responsible for the smaller fruit of MT-*Get* compared to MT because both genotypes have the same MT allele (Figure 5B). Despite the large chromosome 6 segment from *S. galapagense* introgressed into MT-*Get* (Figure 5B), this line has the same red fruit characteristic of MT (Appendix A). This is due to the absence of the *S. galapagense B* allele, which is responsible for orange fruits (Appendix A) and also maps on the long arm of chromosome 6 [42] but, accordingly, outside the introgressed region. Finally, the reduced size of MT-*Get* sepal (Appendix A) is probably an effect of the *EJ-2* allele from *S. galapagense* (Appendix A) since the corresponding MADS-box gene controls the dimensions of the organ in this floral whorl [43].

### 2.3. Whitefly Resistance Test and Trichome Gland Exudation in MT-Get Plants

Since type-IV trichomes drive whitefly (*Bemisia tabaci* Gen.; Hemiptera, Aleyrodidae) resistance in the same accession of *S. galapagense* used here [46,47,48], we examined whether MT-*Get* displayed increased resistance to this insect. MT-*Get* did not differ from MT in a preliminary assay based on the number of whitefly nymphs on leaves after exposure to a controlled greenhouse infested with whiteflies (Appendix A). We also observed that MT-*Get* did not display exudates at the tip of the type-IV trichome gland. The production of such exudates is a feature typical of *S. galapagense* (Appendix A) that accounts for its sticky leaves resulting from AS accumulation [1]. We also noticed that, differently from *S. galapagense*, MT-*Get* leaves were not sticky when touched. *S. galapagense* exudates on trichome glandular heads can be stained with rhodamine B, an AS dye. In MT-*Get*, rhodamine B staining was restricted to the area inside the gland (see insets in Appendix A). These results prompted us to profile the AS accumulated on leaves and the expression of type-IV trichome-specific AS biosynthesis genes [11,12,13] in MT-*Get*.

### 2.4. Acylsugar Accumulation and Related Gene Expression in MT-Get

Since type-IV trichomes are the primary sources of AS [18,19,49], we hypothesized that this compound would accumulate on the leaves of adult plants of the domesticated tomato upon the introgression of genetic determinants leading to the capacity of maintaining the development of type-IV trichomes. On the other hand, the apparent lack of whitefly resistance in MT-*Get* suggested that AS may not have accumulated as much as in *S. galapagense*. We, therefore, conducted a metabolic profile analysis using both liquid chromatography and mass spectrometry to assess AS accumulation in trichome glandular heads of MT-*Get* leaves of adult plants compared to the parental *S. galapagense.*

*S. galapagense* showed peaks corresponding to a variety of sucrose (S)-based AS with different acyl moieties, ranging from 2 to 12 carbons (C2 to C12) (Figure 6, Table 1). Consistently, the AS peaks were very attenuated or absent in the cultivated tomato (MT), which is already known to accumulate very low amounts of AS (Figure 6) [50]. Although MT-*Get* accumulated more AS than MT, it showed statistically significant qualitative and quantitative differences from *S. galapagense*. Notably, MT-*Get* showed reduced levels of AS harboring C10 and C12 moieties, such as S4:23 (2,4,5,12), S4:22 (2,5,5,10), and S4:24 (2,5,5,12) (Figure 6, Table 1). The amounts of S4:23, S4:22, and S4:24 were 120-, 42-, and 18-fold lower in MT-*Get* than *S. galapagense*, respectively (Table 1). These differences are greater than those for type-IV trichome densities between *S. galapagense* and MT-*Get*, which were 3.5-fold higher on the abaxial side and 6.7-fold higher on the adaxial side (Figure 1 and Figure 4). In the GC-MS analysis, detectable peaks of n-decanoate (C10) and n-dodecanoate (C12) were observed only for *S. galapagense* (Appendix A). These carboxylates are derived from C10 and C12 harboring acylsugars, which agrees with the higher amounts of the acylsugars S4:23 (2,4,5,12), S4:22 (2,5,5,10), and S4:24 (2,5,5,12) found in the LC-MS analysis in *S. galapagense* (Figure 6). Only small quantities of methyl dodecanoate were detected in MT-*Get* (Appendix A), which correlates with the presence of S3:22 (5,5,12) and S4:24 (2,5,5,12) in this genotype (Figure 6).

We then evaluated the relative expression of the known genes involved in AS biosynthesis. Four acyltransferases (ASAT) act sequentially to esterify acyl chains in specific positions of the sugar moiety [12,13]. The expression levels of the corresponding four *ASAT* genes were higher in *S. galapagense* leaves than MT-*Get* (Figure 7), which correlates with the differences in AS content (Figure 6, Table 1). The relative expression of the genes coding for acylsugar hydrolase (ASH) enzymes was also quantified (Appendix A). They are responsible for removing acyl chains from specific AS positions, thus creating the substrate for ASAT action [14,51]. The relative expression of *ASH1* in MT-*Get* was significantly higher than *S. galapagense* (Appendix A), which might also reflect the differences in AS content observed between these two genotypes. Since MT-*Get* and *S. galapagense* seem to differ in the capacity to exudate AS (Appendix A), we finally assessed the gene expression of a putative efflux transporter, which may be responsible for AS exudation in type-IV trichome tips. Notably, the ABC transporter (Solyc03g005860), previously associated with AS exudation [52], showed higher expression in *S. galapagense* compared to MT*-Get* (Appendix A).

## 3. Discussion

Type-IV trichomes are involved in critical mechanisms of arthropod herbivory resistance in the *Solanum* genus and beyond. In the wild species *S. pennellii*, *S. galapagense*, and *S. habrochaites*, glandular trichomes are found at high densities, resulting in a high AS accumulation [13,22,26,27,29]. These specialized metabolites protect plants via toxicity and stickiness, thereby trapping and immobilizing the insects or labeling them for predator recognition [20,48,53,54]. Since type-IV trichomes are not found on adult leaves of the cultivated tomato (*S. lycopersicum*) [31], obtaining and studying a tomato genotype with this novel phenotype is a critical step toward a better understanding of broad insect resistance based on AS, as well as elucidating the molecular mechanisms of glandular trichome development. We created such a genotype by introgressing the trait from *S. galapagense* LA1401 into *S. lycopersicum* cv. Micro-Tom, and named the novel introgression line “*Galapagos-enhanced trichomes*” (MT-*Get*).

MT-*Get* displays several intermediate between both parents: it has fewer glandular trichomes than *S. galapagense*, but a higher diversity of trichome types, whereas the comparison with MT has the opposite trend: MT-*Get* has more glandular trichomes and less diversity (Figure 8A). The reduction of the trichome diversity in *S. galapagense* and MT-*Get* is mainly represented by the reduced densities (or absence) of trichomes types III, V, and VI. For type-V trichomes, its inverse correlation with type-IV structures was already reported in a previous study comparing leaves from juvenile and adult phases of tomato cultivars [31]. This result suggests that both trichome types may have an overlapping ontogeny since they only differ anatomically by the presence/absence of a terminal gland [6,7]. The density reduction of trichomes types III and VI might also be a consequence of the increased number of type-IV trichomes via a general trichome initiation and differentiation mechanism that controls the identity of neighboring epidermal cells. The mechanism that prevents trichome formation in clusters is well known in Arabidopsis [55] and was recently also suggested for tomato [54]. Our findings reveal the complexity of studying trichome distribution as a trait in tomato since perturbations in one trichome type are likely to produce changes in the abundance of other types as a pleiotropic effect. Interestingly, the type-IV trichome density was consistently higher on the abaxial side in all conditions in which it was found, such as MT juvenile leaves (Appendix A), MT-*Get* juvenile and adult leaves (Appendix A and Figure 4A,B), and adult leaves of *S. galapagense* and its F_1_ hybrid (Figure 1D,E). It is tempting to speculate that this distribution of type-IV trichomes may have been selected in response to the settling preference of whiteflies [56].

Our analysis further revealed that the development of type-IV trichomes is associated with multiple genes dispersed over several chromosomes. It is not clear yet whether all genomic fragments (loci) identified in this study are essential for type-IV trichome development. However, this result led us to hypothesize that the trait has a polygenic inheritance and that some of the genes within these regions can explain the additional traits found in MT-*Get*. One region found on chromosome 2 coincides with a previously described QTL associated with adult whitefly survival and the presence of type-IV trichomes in segregating populations (F_2_ and F_3_) from a cross between the cultivated tomato and *S. galapagense* [46,48]. This is strong evidence that at least one gene necessary for type-IV trichome development is in this region of chromosome 2. However, the preliminary analysis of sub-lines derived from MT-*Get* harboring *S. galapagense* alleles only in this region suggests that it is not sufficient for high type-IV trichomes densities. A similar conclusion can be drawn for the *S. galapagense* alleles present on chromosomes 1 and 3, whose corresponding sub-lines showed type-IV trichomes but at lower densities. Therefore, it is likely that the high density of type-IV trichomes in MT-*Get* could result from epistatic interactions of different *S. galapagense* alleles. On chromosome 3, MT-*Get* also harbors *S. galapagense* alleles for two known developmental genes: *EJ-2* and *FW3-2*. However, it is unlikely that they can be directly associated with type-IV trichome development since these structures are absent in adult leaves of other species in the tomato clade carrying wild alleles of *EJ-2* and *FW3-2*. This is the case of the *S. cheesmaniae* accessions LA0521 and LA1139, and *S. pimpinellifolium* LA4645 [22,57]. Recently, four QTLs involved in type-IV trichomes development from *S. pimpinellifolium* accession BGV016047 were described [58]. They are located on chromosomes 5, 6, 9, and 11, with the major QTL on chromosome 9. This information suggests that even closely related species could have recruited different genes to control the same trait.

Our initial hypothesis that the presence of type-IV trichome would be sufficient for high AS production and herbivore resistance was not borne out. Our results show that although MT-*Get* plants have a high density of type-IV trichomes, they do not produce AS on the same scale as *S. galapagense* (Figure 8B). Accordingly, MT-*Get* lacked resistance to *B. tabaci*, a major insect pest controlled very efficiently by type-IV trichomes in some wild tomato species [19]. One striking feature was the differential accumulation of AS with medium-chain acyl groups (10 and 12 carbons) between MT-*Get* and *S. galapagense*, suggesting that among the genes controlling the AS metabolic pathway in *S. galapagense*, some could be involved in the esterification of C10 and C12 acyl groups, which may have implications for the levels of insect resistance.

In our analysis of the genes controlling the AS biosynthesis pathway, we noticed *ASAT1* (Solyc12g006330), *ASAT2* (Solyc04g012020), *ASAT3* (Solyc11g067270), and *ASAT4* (Solyc01g105580) are not located within any of the introgressed regions (see Figure 5). This implies that, due to their differential expression, *S. galapagense* and MT-*Get* have distinct alleles for these genes, potentially including *cis*-regulatory elements and transcription factors that are important drivers of domestication [59,60]. Therefore, the effect of both *cis* and *trans* elements on regulating the expression of *ASAT* genes may explain the higher expression of *S. galapagense* compared to MT-*Get*. However, we cannot exclude that the differences in transcript accumulation may also reflect the enriched content of type-IV trichome-derived RNAs in *S. galapagense* due to their higher density than that of MT-*Get* (Figure 1D and Figure 4). On the other hand, the difference in *ASAT4* expression between MT-*Get* and *S. galapagense* is far beyond the magnitude of the trichome density difference between these genotypes. Interestingly, the higher *ASAT2* expression in *S. galapagense* is consistent with the increased levels of C10-12 acyl groups in the wild parental genotype (Figure 6). We propose that the enzyme encoded by *ASAT2* from *S. galapagense* may be able to esterify medium-acyl chains (up to 12 carbons) in the R3 position of the sucrose backbone more efficiently than the MT allele [12] (Figure 7).

Since AS are non-volatile compounds, they are produced in the glands and drip out by a hitherto unclear mechanism [54]. This phenomenon was observed for *S. galapagense* (Appendix A) and may sustain the positive feedback responsible for AS production. A comparative transcriptomic analysis of *S. pennellii* accessions with different AS contents revealed that the expression levels of most AS metabolic genes were positively correlated with AS accumulation [52]. Among the differentially expressed genes (DEGs), three genes putatively encoding ATP-binding cassette (ABC) transporters were upregulated in the accessions with high AS content. Furthermore, an ABC transporter is strongly expressed in the glandular trichomes of *Lavandula angustifolia* (Lamiaceae) [61]. Based on this information, we verified the relative expression of an ABC transporter (Solyc03g005860) in our material and found a similar pattern of *ASAT* expression, i.e., closely correlated to type-IV trichome density on the leaves (Appendix A). While this result suggests that this ABC transporter may be involved in AS exudation, the factors accounting for the differences in AS exudation capacity between *S. galapagense* and MT-*Get* remain to be discovered. It is worth noting that AS transport could be critical in determining how much AS is produced and secreted. One possibility is that AS would remain inside the trichome head without an efficient transmembrane transport mechanism, leading to feedback inhibition of AS production. On the other hand, efficient transport might drive biosynthesis by creating a metabolic flux, potentially preventing feedback inhibition and ultimately leading to high AS accumulation.

We initially expected that type-IV trichomes of *S. galapagense* would have the capacity to accumulate the amounts of sugar moieties necessary to be acylated in the gland tip. Earlier studies using radiolabeled carbon in tobacco showed that isolated trichome glands might be metabolically independent to produce the main exudates, but only when adequately supplied with carbon sources [62]. Subsequent transcription analyses with expressed sequence tags (EST) indicated that trichomes could work with simple biochemical input while having few highly active biochemical pathways in the primary and specialized metabolisms [1]. Although type-IV trichomes contain chloroplasts (Appendix A), these are probably not enough to sustain both primary and specialized metabolism occurring in the cells of this structure [1,63]. Therefore, the differences in AS accumulation are unlikely to be fully explained by genes related to modifications of the acyl moiety, such as *ASATs* and *ASHs*. Furthermore, additional genes involved in sugar metabolism or transport could enable the trichome gland to become a stronger sink.

## 4. Materials and Methods

### 4.1. Plant Material, Growth Conditions, and Breeding Scheme

Seeds of *Solanum galapagense* LA1401 were obtained by the Tomato Genetics Resource Center (TGRC—University of California). Micro-Tom (MT) seeds were donated by Dr. Avram Levy (Weizmann Institute of Science, Israel) and maintained through self-pollination as a true-to-type cultivar since 1998. The “*Galapagos-enhanced trichomes*” (MT-*Get*) line was generated by crossing MT x *S. galapagense* LA1401 using MT as the female donor and the recurrent parent in the further backcrosses (Figure 1). The introgression process followed the procedure previously described by [64], and was based on visual screening on a stereoscope for the presence of a high density of type-IV trichomes on the leaves of adult plants. Considering that the MT cultivar has type-IV trichomes up to the third leaf, we sampled the fifth leaf, which develops during the adult phase of the plant life cycle [31]. Plants were grown in a greenhouse under natural conditions, with a mean temperature of 28 °C, a 11.5 h/13 h photoperiod (winter/summer), and 250–350 μmol photons m^−l^ s^−2^ PAR irradiance attained by reduction of natural radiation with a reflecting mesh (Aluminet, Polysack Industrias Ltd.a., Leme, Brazil). The seeds were sown in 350-mL pots with a commercial potting mix (Basaplant, Artur Nogueira, Brazil) and expanded vermiculite mixture (1:1) supplemented with 1 g L^−1^ NPK (10:10:10) and 4 g L^−1^ dolomite limestone (MgCO_3_ + CaCO_3_). Watering was provided four times a day by an automated bottom irrigation system. After around 12 days, upon the appearance of the first true leaf, the seedlings were transplanted to 150-mL pots containing the soil mix described above with 8 g L^−1^ NPK supplementation. In addition, plants received additional fertilization of 0.5 g NPK (10:10:10) after flowering.

### 4.2. Plant Genetic Transformation

We used a plasmid containing the green fluorescent protein (GFP) reporter driven by the promoter of a gene encoding an acetyl-CoA–dependent acyltransferase enzyme (Solyc01g105580, *SlAT2*). This gene is specifically expressed in the gland cells of trichome types I and IV [13]. The construct was introduced into *Agrobacterium tumefaciens* LBA4404 and used to transform MT and MT-*Get* as described by [64]. Plants regenerated under kanamycin selection were acclimated in a greenhouse and cultivated as described above.

### 4.3. Trichome Morphometry

The methodology used to identify and count trichomes was fully described by [31]. The fifth leaf of each plant was dissected into 15 × 3 mm strips of the leaf blade for the analysis. The samples were affixed to the edge of microscope slides using transparent nail polish. A support made of Styrofoam with a slit was used to secure the slides horizontally but elevated from the stage to keep the trichomes perpendicular to the objective-stage direction. Photographs were taken using a stereomicroscope Leica S8AP0 (Wetzlar, Germany) at 50× magnification coupled to a Leica DFC295 camera (Wetzlar, Germany). The lateral views of the trichomes allowed the correct measurement, classification, and counting of each type. Eight individual plants per genotype were sampled, and four different strips were analyzed per plant for each leaf side (abaxial/adaxial). From the pictures, trichome counting was performed for density estimation. Trichome length measurements were performed using the manufacturer’s analytical software (Leica Application Suite 4.0, Wetzlar, Germany).

### 4.4. Scanning Electron Microscopy

Leaf samples were fixed in Karnovsky solution for 24 h at 4 °C. The material was washed twice with 0.05 M cacodylate solution for 10 min and fixed again in osmium tetroxide (1%) for 1 h. Subsequently, the samples were washed with distilled water and dehydrated in a series of acetone baths. The dehydrated samples were submitted to drying to the critical point and subsequently coated with gold (~3 nm). The observations were performed on an LEO 435 VP scanning electron microscope (SEMTech Solutions, Billerica, MA, USA) operated at an accelerating voltage of 20 kV.

### 4.5. Fluorescence Microscopy

For trichome-specific expression of GFP under the *SlAT2* promoter, analyses were carried out under a Nikon SMZ18 stereoscope attached to a camera (Nikon Corp., Tokyo, Japan). Excitation at 480 nm and a 505 nm emission filter detected fluorescence specifically from GFP. For chloroplast fluorescence detection, trichomes were observed under a Carl Zeiss Axioskop 2 microscope coupled to an Axiocam MRc Zeiss camera using a 540/625 nm excitation/emission filter (Carl Zeiss Microscopy Deutschland GmbH, Oberkochen, Germany).

### 4.6. Mapping-by-Sequencing Analysis

For this experiment, a segregating population (BC_7_F_2_) was produced. The MT-*Get* lineage (BC_6_F_n_) was crossed again with MT, and we waited for one more self-pollination round to carry out trait analysis. The analyzed population was composed of 315 plants that were phenotyped for the presence of type-IV trichomes on leaves of adult plants (i.e., the fifth leaf), as described above. Plants were classified into two categories: with type-IV trichomes (*Get*-like), or without visible type-IV trichomes (MT-like). Five leaf discs (7-mm diameter) were collected from each plant and pooled into two batches according to phenotype before extracting genomic DNA with the method described by [65]. The genomic DNA was further purified with the MasterPure kit (Lucigen, MC85200, Middleton, MI, USA) and submitted to sequencing on a HiSeq PE150bp (Illumina, San Diego, CA, USA) at Novogene (https://en.novogene.com) with 30× depth. Fastq files for each population were concatenated and submitted to quality check by FastQC on the Galaxy platform (https://usegalaxy.org (accessed on March 22, 2018)). Mapping of reads against the cv. Micro-Tom v.1 genome reference sequence (Sol_mic assembly: http://gbf.toulouse.inra.fr/Genomecom (accessed on March 22, 2018)) was carried out with the “Map with BWA for Illumina” (v.1.2.3) software on Galaxy, using default parameters [66] to generate “.sam” files. The variant calling against the cv. Micro-Tom assembly was performed with Samtools Mpileup (v.1.8) [67] and BCFtools call (v.1.6) to generate VCF files. Further comparisons between *Get*-like variants (S1) against the MT-like population (S4) were performed with BCFtools (v.1.6) on the command line interface. The variant allelic frequencies were filtered per site relative to the MT genome sequence using the following parameters: (i) total depth of reading (10 < DP < 100); (ii) allele frequency (AF ≥ 0.8), which is defined as the alternative depth of reading (AD) divided by the total depth of reading (DP); and (iii) number of variants per 1 Mbp window ≥ 30 [36,68]. The analysis pipeline can be seen in Appendix A.

### 4.7. Micro-Tom and S. galapagense Allele Genotyping

Genomic DNA was extracted using the protocol described by [65]. The DNA quantity and quality were determined using agarose gel electrophoresis and a NanoDrop One spectrophotometer (ThermoFisher Scientific, Waltham, MA, USA). The genotyping was performed using CAPS markers that discriminate *Solanum lycopersicum* cv. MT and *Solanum galapagense* alleles (Appendix A). Each 12-μL PCR reaction contained 1.0 μL DNA, 1.2 μL Taq buffer (10×), 1.5 μL MgCl_2_ (25 nM), 0.2 μL dNTPs (10 mM), 0.4 μL each primer (10 pM), 0.1 μL Taq DNA polymerase (5 U μL^−1^—Thermo Fisher Scientific), and 7.2 μL distilled water. The PCR programs were developed according to the optimum annealing temperatures and amplicon sizes of each primer set. The digestion reactions (10 μL) contained 4.0 μL PCR product, 1.0 μL enzyme buffer (10×), 0.2 μL restriction enzyme, and 4.8 μL water. The products were analyzed on 1.5% (*w*/*v*) agarose gels, using SYBR^®^ Gold Nucleic Acid Gel Stain (ThermoFisher Scientific).

### 4.8. Plant Phenotyping

The lengths of the main stem and secondary branches were measured on 48-day-old plants, and the branching index was calculated using [69]. All MT and MT-*Get* plants were grown in 250-mL pots. For fruit mass measurements, many ovaries were pollinated were hand-pollinated with their own pollen. After the fruit set confirmation (five days after pollination), we performed selective fruit thinning to allow only five fruits to develop and ripen on each plant.

### 4.9. Herbivory Test with BEMISIA Tabaci

Seeds were sown in plastic trays using coconut fiber substrate and remained in greenhouse conditions (see the section “Plant material, growth conditions, and breeding scheme”) until transplanting into 8L pots with the substrate. Each pot received five plants of the same genotype (MT or MT-*Get*). Plants were kept in a greenhouse until 23 days after transplanting. During the interval between transplanting and the beginning of infestation with whiteflies (*Bemisia tabaci*), plants received the appropriate cultural treatments and fertilization. After 23 days, the pots were randomly placed in a greenhouse chamber (7 m × 15 m) highly infested with a whitefly population, where the insects were bred and kept exclusively for tomato resistance tests. The pots remained there for seven days to allow eggs to be laid on the leaves. One pot of each genotype containing five plants was randomly sampled after an additional seven days to allow the eggs to hatch. Thirty fully expanded leaflets from the fifth leaf were collected to count hatched nymphs from these plants. Nymphs on each leaflet were counted using the stereomicroscope Leica S8AP0 (Wetzlar, Germany) coupled with a Leica DFC295 camera (Wetzlar, Germany).

### 4.10. Rhodamine B Assay for Acylsugar Staining

Freshly cut whole leaflets of *S. galapagense* and MT-*Get* were submerged in a 0.1% aqueous solution of rhodamine B for one minute. Subsequently, the samples were gently immersed in distilled water four times (serially) to remove the excess dye. The images of stained trichomes were taken as described above in “Trichome Morphometry”.

### 4.11. LC-MS/MS Analysis of Surface Extracts

Semipolar metabolites were extracted by placing two freshly cut tomato leaflets from adult plants (fifth leaf) in a 2-mL reaction tube containing 1 mL of 75% methanol. The tube was vortexed in a standard lab vortex for 60 s at maximum speed. Extracts were filtered using PVDF (0.2 µm) filter plates. The supernatant was transferred to a new tube, centrifuged for 5 min at 18,000× *g*, and placed into a glass vial. The analysis of the extracted metabolites was performed on an LC-MS/MS system composed of an Acquity UPLC (Waters GmbH, Eschborn, Germany) and a TripleTOF 5600 mass spectrometer (SCIEX, Toronto, ON, Canada). For the separation of the analytes, 5 µL of extracts were injected into a Nucleoshell RP 18 column (2.7 µm × 150 mm × 2 mm, Macherey-Nagel GmbH, Düren, Germany). A solvent system was composed of A: 0.3 mM ammonium formate acidified with formic acid at pH 3, and B: acetonitrile. The following gradient was used: 0–2 min: isocratic 95% A, 2–19 min: linear from 95% to 5% A, 19–22 min: isocratic 5% A, 22–22.01 min: linear from 5% A to 95% A, 22.01–24 min: isocratic 95% A. The flow rate was set to 400 µL min^−1^ throughout at 40 °C column temperature. Analyte ionization was performed by electrospray ionization in negative mode with the following parameters: gas 1 = 60 psi, gas 2 = 70 psi, curtain gas = 35 psi, temperature = 600 °C, and ion spray voltage floating of −4500 V. CID fragment spectra were generated in SWATH mode [69] with mass windows of 33 Da and rolling collision energies from −10 to −80 V with a collision energy spread of 15 V. The integration of the peak areas was performed by Multiquant (Version 2.0.2; SCIEX, Toronto, ON, Canada).

### 4.12. GC-MS Acylsugar Quantification

The fifth fully expanded leaves were collected, and the extraction was conducted according to the methodology described by [70]. The compounds were separated via GC-2010 gas chromatography (Shimadzu Corp., Kyoto, Japan) attached to a QP 2010 Plus mass spectrometer (Shimadzu Corp., Kyoto, Japan), using Helium as the charging gas. Hexane was injected into a DB-WAX apolar column (0.25 mm diameter, 30 m length, and 0.25 μm film thickness) to separate the acyl groups from the acylsugar molecules. The data were analyzed using the Lab Solutions-GC/MS software version 2.5 (Shimadzu Corp., Kyoto, Japan). Compound identification was based on the retention time of chromatographic peaks, and the fragments of the mass spectrometer were compared to available standards and data libraries (Wiley^®^ 8 and FFNSC 1.3). The identified compounds were quantified using a calibration curve derived from the peak areas of the standards.

### 4.13. Gene Expression Analyses

The expression analysis of key genes involved in the AS biosynthesis pathway was performed by quantitative real-time PCR (RT-qPCR) in MT, MT-*Get*, and the wild species *S. galapagense*. The comparison was performed only in genotypes harboring type-IV trichome in all leaves. Total RNA was isolated using the mirVana™ Isolation Kit (Ambion, Austin, TX, USA) according to the manufacturer’s instructions. RNA was quantified on a NanoDrop One UV-Vis Spectrophotometer (Thermo Scientific, Waltham, MA, USA), and the RNA integrity was examined by gel electrophoresis. Total RNA was treated with TURBO DNA-free™ Kit (Invitrogen™, Waltham, MA, USA) and subsequently used for cDNA synthesis using the SUPERSCRIPT™ IV 1st Strand Synthesis kit (Invitrogen) according to the manufacturer’s instructions. RT-qPCR reactions were conducted in a 10-µL total volume using a 2× GoTaq^®^ qPCR Master Mix (Promega, Madison, WI, USA), and run on an ABI 7500 qPCR thermocycler (Applied Biosystems). The constitutive housekeeping genes *ACTIN* (Solyc04g011500) and *ELONGATION FACTOR 1 ALPHA* (*EF*-*1α*, Solyc06g005060) were used as internal controls [71,72]. We used three biological replicates, each composed of five leaves, and three technical replicates. The threshold cycle (C_T_) was determined automatically, and the fold changes in expression for each gene of interest were calculated using the equation 2^−∆∆Ct^ [73]. The RT-qPCR primer sequences are listed in Appendix A.

### 4.14. Statistical Analyses

The LC-MS data were converted to the Log10 function before analysis. The statistical comparisons were made using Student’s *t*-test or Tukey’s with GraphPad Prism version 7.00 for Mac (GraphPad Software, La Jolla, CA, USA) and SAS software (SAS Institute Inc., 2013, Cary, NC, USA).

## 5. Conclusions

The results presented here and their implications are summarized in a model in which the transfer of AS-based insect resistance from a wild species to the cultivated tomato requires the stacking of at least three types of genetic determinants: (i) Favorable alleles necessary to build the specific glandular trichomes at the correct developmental stage, such as in MT-*Get*; (ii) Favorable alleles necessary for specific metabolic pathways (e.g., different compositions of acyl groups and capacity to accumulate the sugar moiety for local biosynthesis), and (iii) Favorable alleles necessary to transform glandular trichomes into exudating structures, such as transmembrane transporters (Figure 8C). The novel MT-*Get* introgression line and its sublines described here can be used as a chassis for continuing iterations of genetic introgression or transgenic approaches in knowledge-based breeding programs. In other words, MT-*Get* is the first step to creating an insect-resistant tomato cultivar based on AS production—the plants carry the vessels, but a deeper understanding of trichome biology and biochemistry is needed to fill them up with the appropriate metabolic compounds and allow profuse exudation.

## Figures and Tables

**Figure 1 plants-11-01309-f001:**
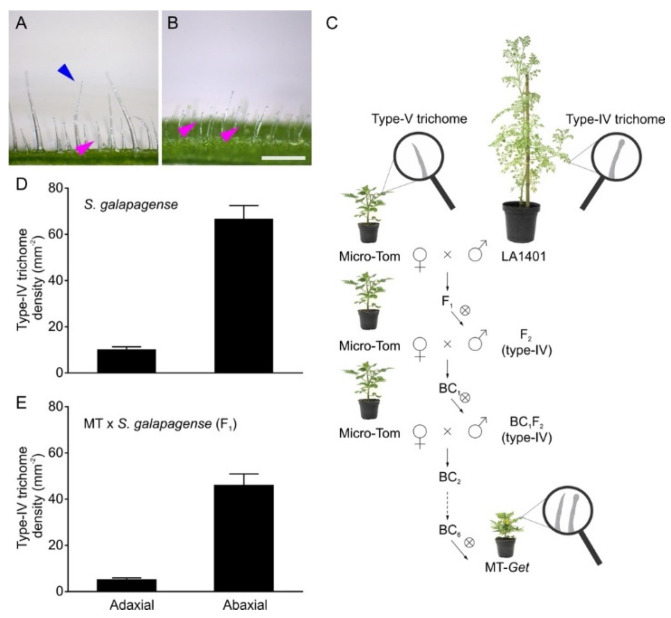
Introgression of *Solanum galapagense* (LA1401) type-IV trichome trait into cultivated tomato (*S. lycopersicum* cv Micro-Tom, MT). Representative light microscopy images showing type-I (blue arrowhead) and type-IV (pink arrowhead) glandular trichomes on the adaxial (**A**) and abaxial (**B**) sides of *S. galapagense* leaf. Scale bar = 250 μm. (**C**) Introgression scheme used to create the Micro-Tom (MT) genotype bearing type-IV trichomes on leaves of the adult developmental phase. The genotype was designated “*Galapagos enhanced trichomes*” (*Get*). ⊗ = self-pollination, BC = backcrossing. (**D**,**E**) Density (mm^−2^) of type-IV trichomes on both leaf sides of the wild species (*n* = 35) and F_1_ plants (MT x *S. galapagense* LA1401) (*n* = 30). Data are means ± SEM.

**Figure 2 plants-11-01309-f002:**
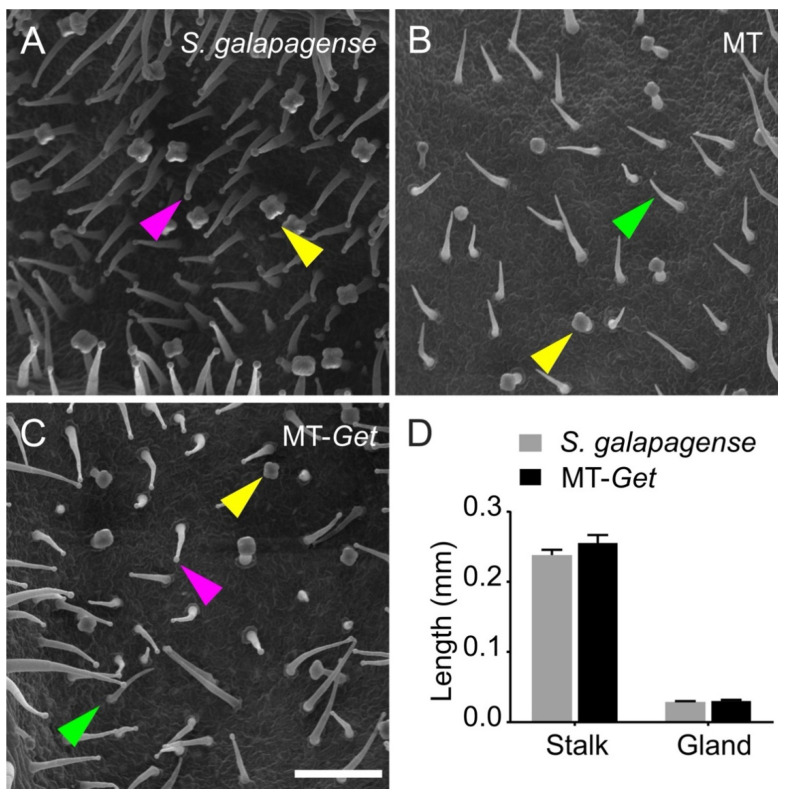
Scanning electron micrographs of abaxial sides of the fifth leaf of representative 25-day-old plants of the tomato wild relative *Solanum galapagense* LA1401 (**A**), Micro-Tom (**B**), and the “*Galapagos enhanced trichomes*” genotype (MT-*Get*) (**C**). Scale bar = 200 µm. The arrowheads point to different trichomes: type IV (pink), type V (green), and type VI (yellow). (**D**) Type-IV trichome stalk and gland length comparisons between *S. galapagense* and MT-*Get*. Data are means (*n* = 30) ± SEM. The data are not statistically different according to Student’s *t*-test (*p* < 0.05 threshold).

**Figure 3 plants-11-01309-f003:**
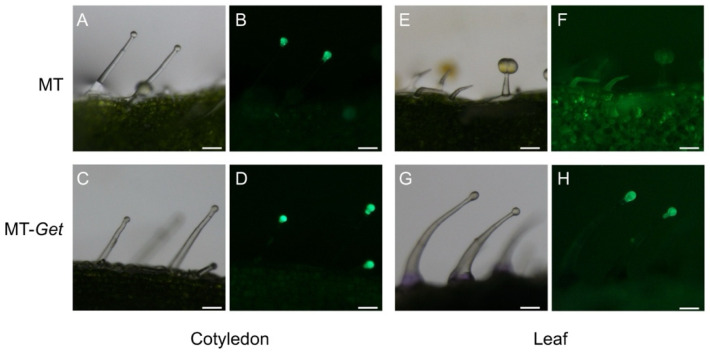
p*SlAT2::GFP* expression (green fluorescence: 480/505 nm excitation/emission spectra) at the tip cells of MT and MT-*Get* type-IV trichomes on the cotyledons (an organ from the juvenile phase) of MT (**A**,**B**) and MT-*Get* (**C**,**D**); and on the fifth leaf (an organ from the adult phase) of MT (**E**,**F**) and MT-*Get* (**G**,**H**). Scale bar = 100 µm.

**Figure 4 plants-11-01309-f004:**
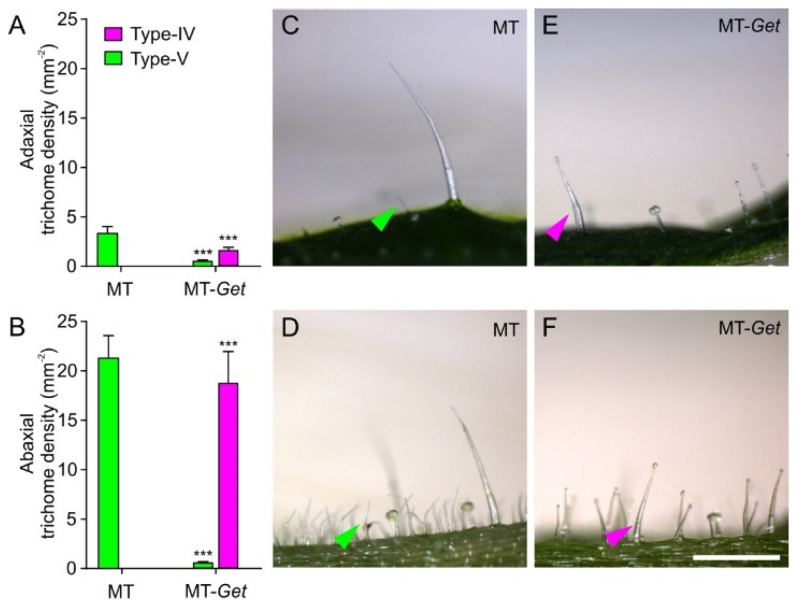
Density (mm^−2^) of type-IV and -V trichomes on the adaxial (**A**) and abaxial (**B**) sides of the fifth leaves from 45-day-old plants of MT and MT-*Get*. (**C**,**D**) Representative micrographs of both sides of the fifth adult leaf from 45-day-old MT plants. (**E**,**F**) Representative micrographs of both sides of the fifth adult leaf from 45-day-old MT-*Get* (**F**) plants. Data are means (*n* = 40) ± SEM. Asterisks indicate a significant difference when compared with the reference sample according to the Student’s *t*-test at *p* < 0.001. (***) Scale bars = 250 μm.

**Figure 5 plants-11-01309-f005:**
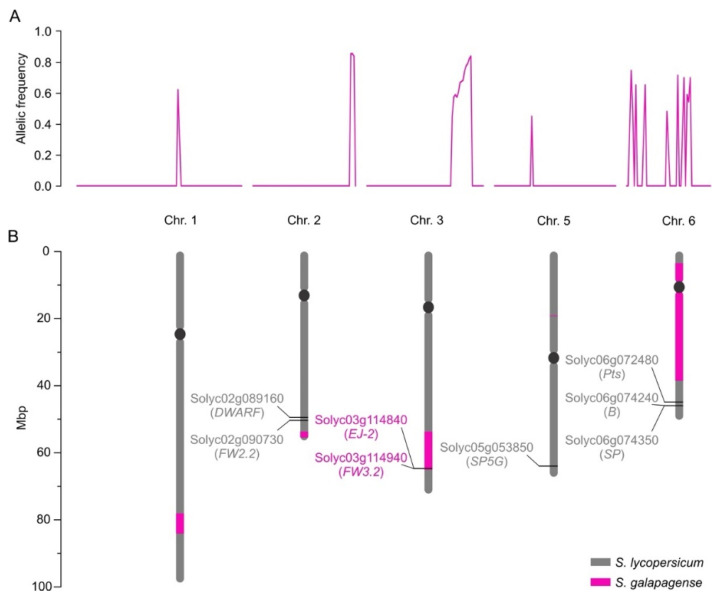
Mapping-by-sequencing results. (**A**) Allelic frequency indicates the chromosomal positions where MT-*Get* has introgressions from *S. galapagense* LA1401. (**B**) Representation of the corresponding positions of the chromosomal fragments from *S. galapagense* (pink bars) introgressed into MT-*Get*. The positions were based on the *S. lycopersicum* cv. Heinz reference genome sequence. Relevant genes (as discussed in the text) and their SGN (Solyc) ID numbers are represented. Note that the MT-*Get* genome bears the MT-mutated alleles for *DWARF* [37] and *SP* [38], which are determinants of the MT reduced plant size and determinate growth habit, respectively [39]. The presence of the MT allele at the *SP5G* locus also contributes to the reduced plant size of MT-*Get*, since the *S. galapagense* allele promotes additional vegetative growth due to a lack of flower induction under long days [40]. MT-*Get* also lacks *S. galapagense* alleles for genes conferring additional phenotypes distinct to this wild species, such as highly dissected leaves (*Pts*) [41], and β-carotene accumulating fruits (*B*) [42]. Appendix A depicts and discusses the effect of the wild species alleles *EJ-2* and *FW3.2* on the MT-*Get* phenotype.

**Figure 6 plants-11-01309-f006:**
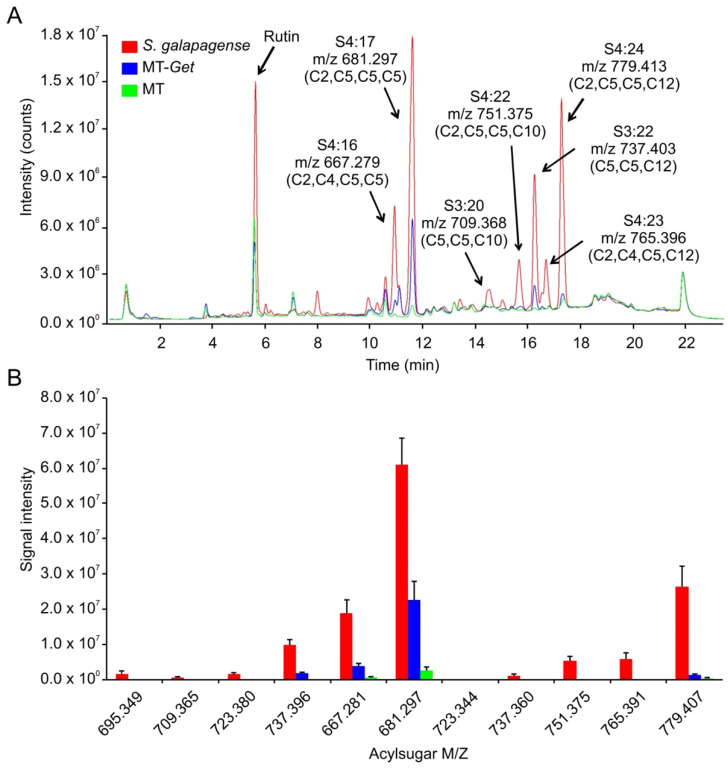
Acylsugar (AS) content in *Solanum galapagense* LA1401, Micro-Tom (MT), and MT-*Get*. (**A**) Representative LC-MS/MS chromatogram. Peak area quantifications and statics are shown in Table 1. (**B**) Signal intensity for each of the AS analyzed in the three genotypes. The symbols used to classify acylsugars are detailed in the Introduction section.

**Figure 7 plants-11-01309-f007:**
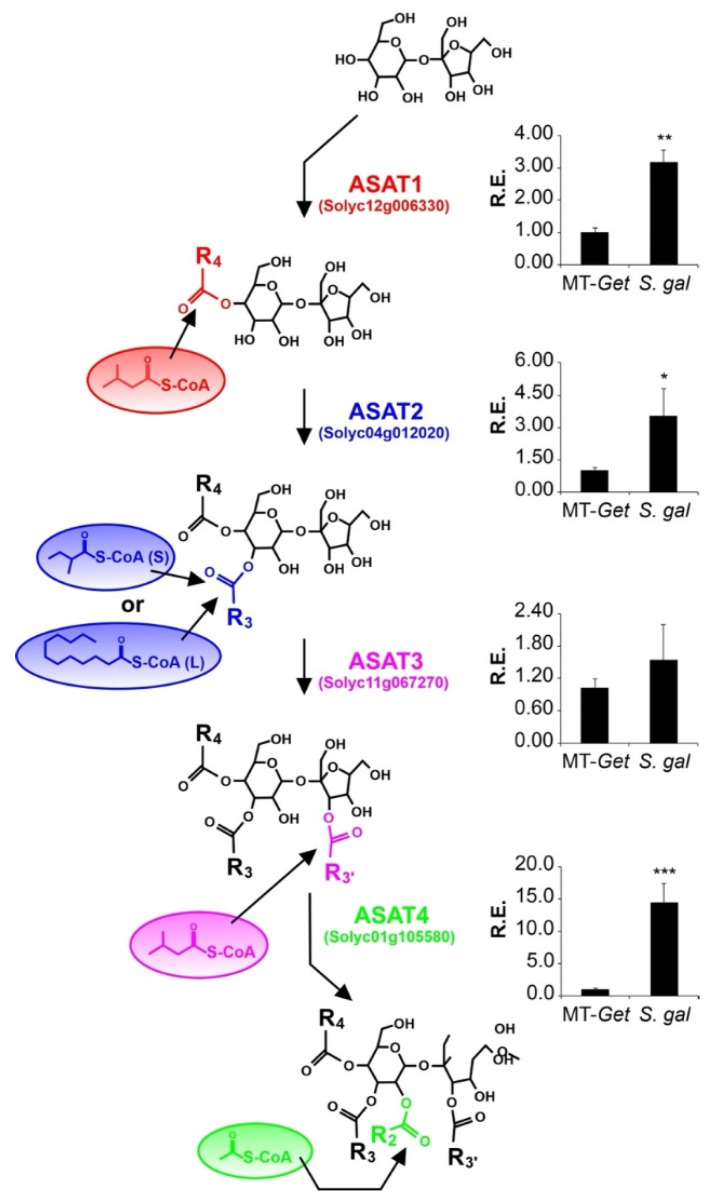
Relative expression of acylsugar acyltransferase (*ASAT*) genes in leaf tissues of MT-*Get* and *S. galapagense*. RT-qPCR values are means ± SE. Biological triplicates were averaged and analyzed statically according to the Student’s *t*-test at *p* < 0.05 (*); *p* < 0.01 (**); *p* < 0.001 (***).

**Figure 8 plants-11-01309-f008:**
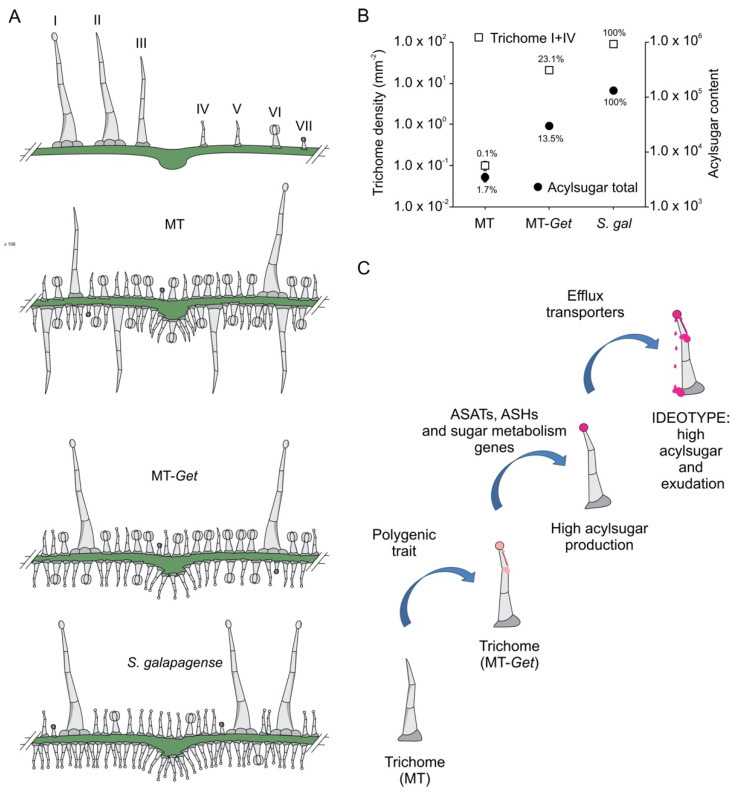
Schematic representation of trichome type distribution in each genotype used in this work (**A**). (**B**) Logarithmic scale representing total trichomes (types I and IV) versus total acylsugar content (signal intensity). (**C**) Sequential steps necessary to obtain broad and durable insect-resistant tomatoes through the high production of acylsugars on tomato leaves. ASAT = acylsugar acyltransferase, ASH = acylsugar hydrolase.

**Table 1 plants-11-01309-t001:** Peak areas from ion chromatograms (LC-MS/MS) of acylsugars from *S. galapagense*, MT-*Get*, and MT.

Predicted AS	*m*/*z*		Peak Area		***S. galapagense***/ MT-Get
		*S. galapagense*	MT-*Get*	MT	
S3:20 (5,5,10)	709.368	675.914 **a**	72.873 **b**	7.192 **c**	9.28
S3:22 (5,5,12)	737.396	9877.349 **a**	1767.007 **b**	208.375 **c**	5.59
S4:16 (2,4,5,5)	667.282	18,907.925 **a**	3835.183 **b**	514.101 **c**	4.93
S4:17 (2,5,5,5)	681.297	61,140.597 **a**	22,592.369 **b**	2509.929 **c**	2.71
S4:22 (2,5,5,10)	751.375	5363.138 **a**	125.538 **b**	26.872 **c**	42.72
S4:23 (2,4,5,12)	765.391	5864.023 **a**	48.632 **b**	26.809 **c**	120.58
S4:24 (2,5,5,12)	779.407	26,335.227 **a**	1446.972 **b**	301.079 **c**	18.20

Acylsugars were identified according to their *m*/*z* and retention time (*n* = 4). For the sake of simplicity, values were divided by 1000. Values followed by different letters in each row are statistically different according to the Tukey’s test (*p* < 0.01).

## Data Availability

DNA-Seq raw dataset used for mapping-by-sequencing was submitted to the NCBI Sequence Read Archive (BioProject # PRJNA776853). All relevant data are within the paper and its Appendix A.

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
