# Peer review of "The Genetic Complexity of Type-IV Trichome Development Reveals the Steps towards an Insect-Resistant Tomato"

_plants, 2022, doi:10.3390/plants11101309_

Round 1
Reviewer 1 Report
Does the domestication of tomato plants resulted in a changed resistency towards insects?
Yes, the domesticated tomato plants lost their resitency towards insects and this might be connected to the loss of the type-IV trichomes which produce acylsugars during tomato domestication.
Will engineered domesticated tomato plants with the type-IV trichomes regain the insect resistance?
No - authors produced tomato plants with the type-IV trichomes. Nevertheless, amount of acylsugars was not enough to get them resistant to whiteflies.
This is original and important work suggesting that other features, besides the type-IV trichomes, have been lost during the process of tomato domestication.
Yes, their conclusions are consistent with the evidence and arguments.
And yes again, they have addressed the main questions posed.
This is highly relevant and carefully performed study reporting new important findings improving significantly our understanding of insect resistance in tomato.
Author Response
Thank you for your careful analysis of our manuscript and encouraging words. Indeed, we are excited about this project and continue working towards revealing the mechanisms behind trichome development in tomato. We appreciate your comments and your careful reading of our work.
Reviewer 2 Report
The manuscript was well written. I only suggested minor changes. The title may still needs work. Hope my suggested changes make sense. Hope other reviews will have good insights for the title as well.

Author Response
Thank you for your careful reading of our work and suggestions to improve our manuscript. We considered each suggestion very carefully and appreciate your insights. In line 317, you asked whether the resistance conferred by acylsugars is just against insects or not. In fact, this type of resistance is against arthropod herbivory in general. We have edited that sentence to make it clear. Thank you for pointing this out to us. Regarding the title, we appreciate your suggestion. The suggestion of replacing “informs” with “reveals” improved the title - thank you! On the other hand, we prefer to avoid the word “mechanisms” because we have not yet identified the culprit genes that are involved in type-IV trichome development. Our work, however, reveals that multiple loci are involved in this process and we present discrete chromosomal segments that are currently under investigation in our lab to identify and characterize these genes.